# High-throughput, image-based phenotyping reveals nutrient-dependent growth facilitation in a grass-legume mixture

Kirsten Rae Ball [1,2¤]*, Sally Anne Power[1], Chris Brien[3], Sarah Woodin[2], Nathaniel Jewell[3], Bettina Berger[3], Elise Pendall[1]

**1** Hawkesbury Institute for the Environment, Western Sydney University, Penrith, New South Wales, Australia, **2** Institute of Biological & Environmental Sciences, University of Aberdeen, Aberdeen, United Kingdom, **3** Australian Plant Phenomics Facility, The Plant Accelerator, School of Agriculture, Food and Wine, University of Adelaide, Urrbrae, South Australia, Australia

¤ Current address: Environmental Sciences, University of Arizona, Tucson, Arizona, United States of America.
* kirsten.ball@email.arizona.edu

**Data Availability Statement:** Data are available via Figshare. DOI: (10.25909/12895121).

**Funding:** The Australian Plant Phenomics Facility received grant funding from the Australian

## Abstract

This study used high throughput, image-based phenotyping (HTP) to distinguish growth patterns, detect facilitation and interpret variations to nutrient uptake in a model mixed-pasture system in response to factorial low and high nitrogen (N) and phosphorus (P) application. HTP has not previously been used to examine pasture species in mixture. We used red-green-blue (RGB) imaging to obtain smoothed projected shoot area (sPSA) to predict absolute growth (AG) up to 70 days after planting (sPSA, DAP 70), to identify variation in relative growth rates (RGR, DAP 35–70) and detect overyielding (an increase in yield in mixture compared with monoculture, indicating facilitation) in a grass-legume model pasture. Finally, using principal components analysis we interpreted between species changes to HTP-derived temporal growth dynamics and nutrient uptake in mixtures and monocultures. Overyielding was detected in all treatments and was driven by both grass and legume. Our data supported expectations of more rapid grass growth and augmented nutrient uptake in the presence of a legume. Legumes grew more slowly in mixture and where growth became more reliant on soil P. Relative growth rate in grass was strongly associated with shoot N concentration, whereas legume RGR was not strongly associated with shoot nutrients. High throughput, image-based phenotyping was a useful tool to quantify growth trait variation between contrasting species and to this end is highly useful in understanding nutrient-yield relationships in mixed pasture cultivations.

## Introduction

Farm management plays a crucial role in global food security as it affects food production and the environmental impacts of agricultural practices [1]. Generally, fertilisation increases crop yield; however, mineral fertilisers are costly and can have detrimental effects on the wider

Government through the National Collaborative Research Infrastructure Strategy (NCRIS). KB received a Postgraduate Internship Award from the Australian Plant Phenomics Facility towards the completion of this project".

**Competing interests:** The authors have declared that no competing interests exist.

environment [2]. Nitrogen (N) fertilisation can accelerate soil carbon turnover and gaseous emissions [3, 4], and assuming zero growth in agricultural production, world rock phosphate (P) reserves are only expected to last approximately 260 years [5]. As global grain demand is projected to increase by around 50% in the next 30 years, environmental impacts from fertiliser applications will become unsustainable, highlighting the need to improve agricultural nutrient use efficiency [1, 5–7].

Optimising crop growth under nutrient application can be achieved *via* several different practices. These include preferential planting of 'environmentally appropriate' species [8], precision nutrient application to match plant demand [9], genetic engineering [10] and intercropping of complementary species [11]. Plants have two overarching growth strategies in response to N and P availability: (i) increasing nutrient uptake and directing them towards greater or more rapid growth, or (ii) conserving and storing nutrients and slowing or maintaining growth-rate [12, 13]. Plant growth strategies involve trade-offs between increasing both productivity and resource acquisition (interpreted herein as an acquisitive growth strategy), and reducing productivity and conserving resources (interpreted herein as a conservative growth strategy) [12]. Grasses require substantial amounts of mineral N, mainly owing to rapid growth rates and a lengthy growing season [14], and legumes are slower growing and less reliant on mineral fertilisers [15, 16]. Legumes can provide N to other species *via* a symbiotic relationship with nitrogen fixing bacteria but require moderately high levels of P to support biological nitrogen fixation (BNF) [17, 18]. Ultimately, plant growth strategies will determine growth rates, nutrient requirements and interactions with other plants [12], and species growth strategies drive variations to primary productivity in agricultural systems [19].

Interspecific interactions between plants can be *negative*, in the case of competition for resources [20], *neutral*, where complementarity ensures that species do not compete for the same resources [11], and *positive*, where facilitation leads to higher performance of a species when grown in mixture than in monoculture [21]. In mixed species pastures, niche complementarity and facilitative relationships can be beneficial. If facilitation is occurring, the total production of a species is likely to be significantly greater in a mixture than its average production in a monoculture—referred to as "overyielding" [22–24]. Mechanisms behind overyielding have been attributed to belowground processes such as enhanced mobilisation of soil nutrients [25] and aboveground interactions including microclimate improvement and light partitioning [26]. Measurements of overyielding combined with examinations of temporal plant growth dynamics can assist us to determine how net primary productivity is likely to vary under particular environmental conditions [27, 28]. The most traditional assessment of plant growth–absolute growth (AG)—reports the total increase in biomass per unit of time and is usually obtained using final harvest weights [29]. Relative growth rate (RGR), a lesser used but potentially more informative assessment of growth, describes the increase in size relative to the size of the plant at the beginning of a time interval [29]. RGR is strongly determined by a plants metabolic requirement for N and P [30], and hence analyses of growth rate diversity in pasture mixtures can assist in detecting potential for nutrient competition or assessing facilitation.

Calculating growth dynamics can be labour- and cost-intensive and often impractical when reliant on destructive harvest techniques [31, 32]. Additionally, calculations of relative growth obtained from weight and height measurements of different species may introduce bias and inaccurate comparison [33, 34]. The recent emergence of high throughput (HT) imaging techniques that allow rapid temporal assessment of plant phenotype by environment interactions, and are scalable to the farm scale, are contributing significantly to relieving this research bottleneck [35, 36]. High throughput phenotyping (HTP) is an automated technique providing quantification of plant traits without the need for destructive harvest [37]. In multiple-image

based systems, cameras operating at different angles allow derivation of a mathematical relationship between several two-dimensional (2D) red-green-blue (RGB) images to quantify plant size [38]. Shoot architectural traits such as canopy biomass, density and leaf area are primarily extracted from RGB images and are used to calculate AG and RGR. Additional information can be acquired using hyperspectral imaging to calculate traits related to foliar nutrient content [25, 39, 40]. All HTP methods require calibration to improve our ability to relate image information to plant growth dynamics, in order that these datasets may be used to measure phenotypic variation in biological systems of interest [25].

This study used image-based HTP to 1) predict absolute and relative growth rates, and from those predictions detect facilitation in a mixed pasture cultivation; and 2) interpret variations in RGR and nutrient uptake between grass-legume cultivations under contrasting N and P inputs (factorial combination of low (L) and high (H) inputs of N and P). As the general growth strategies and responses of grass and legume species to fertilisation and intercropping are well known, we applied HTP to test the following:

1. RGR, AG and nutrient uptake in grasses would be primarily influenced by increased N availability owing to their rapid, nutrient acquisitive growth strategy. RGR and AG in legumes would be preferentially influenced by P availability as P is required in biological nitrogen fixation. Legumes are also expected to display a slower, more nutrient conservative growth strategy and their growth is expected to be less tightly correlated with soil and shoot nutrients.

2. Overyielding (greater yield in mixture compared to monoculture, indicating facilitation) would be present if N or P, but not both, were limiting. This is because P is required by legumes to support biological nitrogen fixation (BNF) and induce facilitation, and under N replete conditions grasses are expected to grow faster.

3. Where overyielding is occurring, because of their slow, conservative growth strategy legume RGR is expected to decline, but their shoot N and P concentrations would remain stable.

## Materials and methods

### Experimental design and species

Our experiment was conducted between the 24th July 2018 and 5th October 2018 at the Australian Plant Phenomics Facility at the University of Adelaide (-34.971298, 138.639627) in a Lemnatec Imaging System, during which time we obtained plant images on 43 consecutive days. The experiment investigated the effects of four fertiliser treatments derived by factorial combination of two levels of nitrogen (LN and HN) and two levels of phosphorus (LP and HP) on one grass (*Phalaris aquatica*) and one legume (*Trifolium vesiculosum*) pasture species grown in monoculture and mixture. The twelve treatments were arranged in a latinized, resolved incomplete-block design with ten replicates, for a total of 120 pots. The two Australian pasture cultivars, Holdfast GT (*P. aquatica*) and Cefalu (*T. vesiculosum*), were chosen as they have a similar upright growth form, are both winter and early spring active and are suited to a wide range of sandy clay loam soils. Seeds were obtained from Heritage Seeds Australia, along with the appropriate group C rhizobial inoculant required for *T. vesiculosum*. The experimental design was obtained using CycDesigN [41] and randomized using the dae package in R [42, 43] (Fig 1).

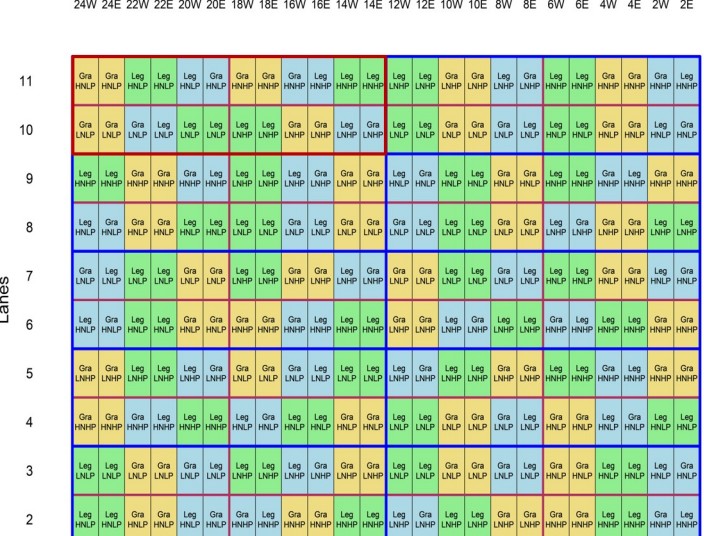

**Fig 1. Main experimental design.** Within each replicate (indicated by the solid red box) the four nutrient combinations (LNLP, HNHP, HNLP and LNHP) were assigned to 4 main units arranged in a grid according to the experimental design. The three species combinations [grass only (Gra), legume only (Leg) and mixture (Gra Leg)] were randomized to the three consecutive pots within each main unit. Coloured blocks indicate 1 pot containing two halves.

## Growing conditions

We used pasteurized, unfertilised potting mixture consisting of pre-mixed (0.33:0.33:0.33 by volume) sand, clay loam and coco peat at pH 6.3. Three kg (dry weight) of soil was potted into 198 mm diameter x 149 mm high (4587 cm$^3$) drainage-drilled pots, each seated on a 200 mm round dish ensuring no water or nutrient loss occurred from the system. Eight seeds were planted in each pot (Day after planting; DAP 0), with two seedlings on each half of the pot after thinning (DAP 16). In mixtures, one half comprised grass while the other half comprised legume. The plants were physically separated aboveground by a white plastic divider, set 5 cm into the soil and oriented north–south on the conveyor system in the greenhouse for consistency with respect to imaging orientation and solar exposure. The divider allowed intermingling of roots belowground (Fig 2).

To promote microbial activity, we added a field soil microbial wash generated from unfertilised (>30 years) old-field pasture soil obtained from the Hawkesbury Forest Experiment in Richmond, NSW (-33.611672, 150.740172). Briefly, 1 g of field soil was mixed with 100 ml of DI water and molasses in a ratio of 100:1 (ml) with the recommended amount of Group C rhizobial inoculant to obtain ~ 3,000,000 rhizobia. The 100 ml of microbial wash was added to every experimental pot on DAP 16 after the seedlings had established. Effective nodulation and belowground biomass were assessed on DAP 35 prior to fertilization in non-experimental pots (n = 6) and both were considered adequate to assert that the inoculation was successful [44, 45] and intermingling of roots was occurring. Ongoing, plants were watered once daily, and soil water content was maintained at field capacity (22% (w/w) gravimetric water content) by watering to weight. Commencing DAP 29, weighing and watering-to-weight was incorporated into the imaging sessions. Twice-daily watering was performed near the end of the experiment (DAP 66–69), when water consumption was high. The greenhouse maintained average

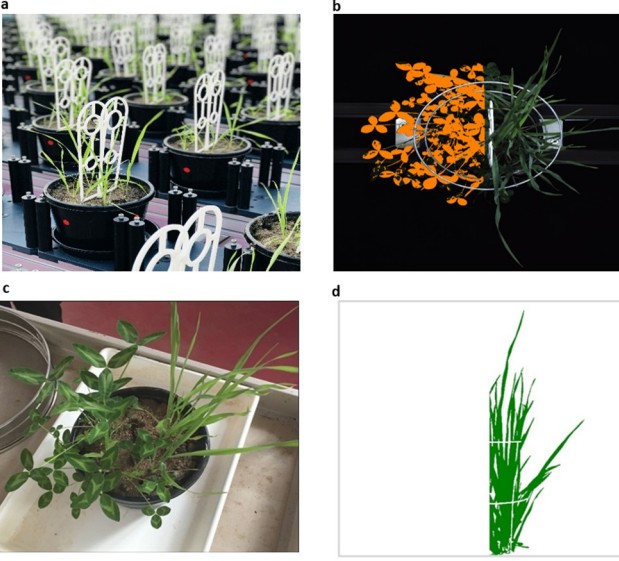

**Fig 2. Pot-level separation of plant biomass and experimental design to improve image differentiation. a)** Cart layout and pot level separation using above-ground dividers (DAP 35); **b)** digitally enhanced RGB image (top view) of a mixed-species pot, showing centre divider (white), grass (natural green colour, right; eastern half of pot) and legume (orange highlights, left; western half of pot); **c)** oblique view of a mixed-species pot at final harvest (DAP 70) with centre divider removed; **d)** digitally processed RGB image (side view) of a mixed-species pot, showing only the grass component in the eastern half (artificial green; superimposed white lines correspond to the enclosing wire frame, which was painted blue for maximum visual contrast).

temperatures of 27°C/16°C on a day/night cycle and an average day length between 07:00–19:00 under natural light conditions for a total of 70 days.

## Fertilisation treatments

Nitrogen (N) and phosphorus (P) were prepared in DI water in the forms of ammonium nitrate ($NH_4NO_3$) and a pH 6.3 balanced mixture of disodium phosphate ($Na_2HPO_4$) and sodium dihydrogen phosphate ($NaH_2PO_4$). Nutrients were added on a dry-weight, mg kg$^{-1}$ of soil basis. The low N-low P (LNLP) treatment nutrients (33 mg N, and 11 mg P, N:P ratio; 3:1) were added to all pots prior to imaging on DAP 16. On DAP 35, nutrients were added to increase the total amount of N and/or P to desired treatment levels. In total, for the high N-high P (HNHP) treatment, we added 99 mg N and 33 mg P (N:P ratio 3:1). For the HNLP treatment, we added 99 mg N and 11 mg P (N:P ratio 9:1). For the LNHP treatment, (33 mg N and 33 mg P, N:P ratio; 1:1). All pots received other macro- and micronutrients at the following rates (mg kg$^{-1}$ dry soil): $K_2SO_4$, (75); $CaCl_2$ $2H_2O$, (75); $MgSO_4$ $7H_2O$, (45); $CuSO_4$ $5H_2O$, (2.1); $ZnSO_4$ $7H_2O$, (5.4); $MnSO_4$ $H_2O$, (6.4); $CoCl_2$ $6H_2O$, (0.33); $Na_2MoO_4$ $2H_2O$, (0.18); $H_3BO_3$, 0.3 and FeEDTA, (0.4). Plant available macronutrients measured in plant-free pots are detailed in S1 File.

## Biomass harvest

On DAP 70, all above ground shoot biomass (AGB) was harvested and soil samples collected at the pot level to determine total AGB, shoot nutrient concentrations and soil extractable nutrients (N and P). AGB was assessed by carefully separating the plants between the two sides of the pot and cutting at soil level to separate from belowground biomass. AGB was dried at 70°C and weighed and the reported AGB is the total weight of the two individual plants from

each side of the pot, here forward referred to as "half-pot". Where "whole-pot" values are reported, this represents the total dry weight of all plant biomass within one pot. All plants were deemed viable for harvest, except a single pot in the grass only LNLP treatment which did not survive. Clerical error was suspected for one half-pot dry-weight observation in the LNLP grass monoculture and LNHP grass monoculture and these observations were removed from analysis. The final dataset included n = 236 useable half-pot observations. All nutrient analysis observations for these replicates were also removed.

## Soil and shoot nutrient analyses

Soil samples were extracted within 12 h of biomass harvest. Extractable N in soil was determined by shaking 40 ml of 2 M potassium chloride (KCl) solution with 4.0 g soil (< 2 mm) at 170 rpm for 1 hour and then filtering through a 2.5 μm ashless filter (Grade 42, Whatman PLC, Kent, U.K). Extractable P was determined by mixing 4 g soil in 40 ml of 0.5 M NaHCO$_3$ and shaking for 16 hours [46]. Soil extracts were stored at -20˚C until colorimetric analysis in a discrete analyser (AQ2, SEAL Analytical, Ltd., Milwaukee, WI USA and EPA135 method). For total carbon (C) and N shoot nutrients, a subsample (~3 g) of AGB from each plant was finely ground and homogenised with an MM 400 mixer mill (Retsch, GmbH, Haan, Germany) and an approximately 5 mg subsample was taken for combustion analysis. C and N concentrations were estimated using an Elementar Vario El Cube Carbon/Nitrogen analyser (Elementar Analysersysteme GmbH, Langenselbold, Germany). Phosphorus concentration of foliar samples was obtained after digesting approximately 55 mg of plant material in concentrated H$_2$SO$_4$ and H$_2$O$_2$ in a microwave digester, and colorimetric analysis following an ammonium molybdate reaction [47]. Measurement error was suspected in two replicates for total C, N and P; one in the HNLP legume monoculture and one in the LNHP grass monoculture and were removed from the analysis.

## Imaging

Each pot was individually imaged daily using four different cameras, comprising one top view camera and three side view cameras at different angles. Of the four images obtained, three (top view and two side views) were used for image analysis and estimation of projected shoot area (PSA). These images had the right orientation to separate the images along the midline, enabling a separation of the two halves of the pot along the plastic divider (Fig 2). LemnaGrid software (LemnaTec GmbH, Aachen, Germany) was used to separate the images into two halves and measure the pixels corresponding to plants in each half. For this, a nearest neighbor colour classification was used to separate foreground and background, followed by noise removal steps. It was decided not to exclude any imaging data as outliers for analysis purposes.

## Data processing and growth calculations from image-based data

Image-based data were processed using the multi-step method of smoothing and extraction of traits (SET) described by [48] with the aid of growthPheno [49], an R package [43]. Firstly, the half-pot projected shoot area (PSA) was defined as the sum of plant pixels visible in the three half-images in each half-pot on each imaging day, yielding *n* = 240 observations per day. The whole-pot projected shoot area (PSA) was defined as the sum of plant pixels visible in the six half-images in each whole-pot on each imaging day, yielding *n* = 120 observations per day. In this paper, the whole-pot data was only used to compare between measured AGB *vs* PSA predicted values. The raw PSA (kpixels) data exhibited a high degree of day-to-day variation and so the next step was to smooth the PSA data to produce what is herein referred to as sPSA; natural cubic smoothing splines with df set to 5 were used (moderate smoothing) [50]. Thirdly,

the time points DAP 35, 40, 50, 60 and 70 were chosen for further analysis based on changes in growth pattern observed from sPSA plots. Only measurements taken after application of nutrients (DAP 35–70) are presented. Finally, RGR (day$^{-1}$) was calculated for the intervals between successive pairs of time points using Eq 1:

$$RGR(t_1, t_2) \; = \; \log[sPSA(t_2) \, / \, sPSA(t_1)] \tag{1}$$

where $t_1$ and $t_2$ are the DAPs defining an interval.

We calculated image-derived AG and overyield from sPSA on DAP 70 and related these values to DAP 70 harvested AGB using correlation analyses.

## Correlations of image-based biomass estimates to harvested aboveground biomass

We used the 'lm' function from Base R [43] to obtain correlation coefficients between our absolute growth values (sPSA DAP 70) and harvested dry weight biomass (AGB) in grams. All model assumptions of normality and homoscedasticity were met.

## Statistical analysis of growth and nutrient concentrations

To obtain growth predictions from the image-based data, and to compare shoot nutrient concentrations between treatments a mixed-model analysis was performed using the R package ASReml-R [51] and asremlPlus [52] packages for the R statistical computing environment [46]. We analysed the half-pot responses based on a maximal mixed model including terms for the treatment differences, spatial effects and residual error variation. The model was of the following form;

$$\mathbf{y} \; = \; \mathbf{X}\boldsymbol{\beta} + \mathbf{Zu} + \mathbf{e}$$

where $\mathbf{y}$ is the response vector of values for the trait being analysed; $\boldsymbol{\beta}$ is the vector of fixed effects; $\mathbf{u}$ is the vector of random effects; and $\mathbf{e}$ is the vector of residual effects. $\mathbf{X}$ and $\mathbf{Z}$ are the design matrices corresponding to $\boldsymbol{\beta}$ and $\mathbf{u}$ respectively. The fixed-effect vector $\boldsymbol{\beta}$ was partitioned as

$$\boldsymbol{\beta}^\top \; = \; [\, \mu \; \boldsymbol{\beta}_R^\top \; \boldsymbol{\beta}_{Si}^\top \; \boldsymbol{\beta}_H^\top \; \boldsymbol{\beta}_{Sp}^\top \; \boldsymbol{\beta}_G^\top \; \boldsymbol{\beta}_L^\top \,]$$

where $\mu$ is the overall mean and the first three $\boldsymbol{\beta}$ subvectors correspond to the effects of replicates (R), greenhouse sides (Si, east or west) and pot halves (H, east or west) that capture systematic spatial variation within the greenhouse; $\boldsymbol{\beta}_{Sp}^\top$ incorporates the two species main effects; $\boldsymbol{\beta}_G^\top$ contains parameters for the 3 main effects, the 3 two-factor interactions and the three-factor interaction of the factors cultivation type, nitrogen and phosphorus for the grass (G) species; and $\boldsymbol{\beta}_L^\top$ contains the same parameters for the legume (L) species. The random-effects vector $\mathbf{u}$ was partitioned as $[\mathbf{u}_{R:M}\mathbf{u}_{R:M:P}]$where $\mathbf{u}_{R:M}$ is the vector of main-unit (M) random effects within each replicate (R) and $\mathbf{u}_{R:M:P}$ is the vector of random effects for pots (P) within each main-unit (M); these vectors captured any non-trend spatial variation. The residuals $\mathbf{e}$ were assumed to be normally distributed with their variance allowed to vary with both species and nitrogen. If the data $\mathbf{y}$ are ordered by species (labels G and L) followed by fertiliser level (labels L and H) and then observations with the combinations of species and fertiliser level,

then the residuals are modelled as:

$$
N\left(\mathbf{0}_{240}, \begin{bmatrix} \sigma_{GL}^2\mathbf{I}_{60} & \mathbf{0}_{60} & \mathbf{0}_{60} & \mathbf{0}_{60} \\ \mathbf{0}_{60} & \sigma_{GH}^2\mathbf{I}_{60} & \mathbf{0}_{60} & \mathbf{0}_{60} \\ \mathbf{0}_{60} & \mathbf{0}_{60} & \sigma_{LL}^2\mathbf{I}_{60} & \mathbf{0}_{60} \\ \mathbf{0}_{60} & \mathbf{0}_{60} & \mathbf{0}_{60} & \sigma_{LH}^2\mathbf{I}_{60} \end{bmatrix}\right),
$$

where $\mathbf{I}_{60}$ and $\mathbf{0}_{60}$ denote identity and zero matrices respectively. For each trait, residual likelihood ratio tests with $\alpha = 0.05$ were used to determine whether the variance model can be simplified by removal of the nitrogen level variance difference and/or species variance difference. The model was modified to reflect the results of these tests and residual-versus-fitted values plots and normal probability plots confirmed that model assumptions were met. Wald F-tests with $\alpha = 0.05$ were conducted for the fixed effects within each species to determine a model for describing how cultivation, nitrogen and phosphorus affect the response for each species. Testing began with the three-factor interaction for a species and, only if it was not significant, proceeded to test the two-factor interactions; the main effects were only tested if that factor had not occurred in a significant interaction. Thus, the estimated marginal means (Lenth et al., 2019), the means for the selected model, were obtained using the R packages ASReml-R [51] and asremlPlus [52].

## Calculating overyielding from sPSA DAP 70 values

Overyielding calculations in mixtures were conducted using methods described by [22, 24]. All assumptions including ensuring consistent planting densities and the allowance of sufficient time for below-ground community interactions to develop, were met [53]. Using our absolute growth (sPSA DAP 70) values we obtained relative yield totals (RYT, Eq 2) for each treatment from the sum of the relative yield (RY) of each species ($i$):

$$
\text{RYT} = \sum_{i=1}^{s} RYi \tag{2}
$$

where $s$ is the number of species $i$ and RY is

$$
RYi = \frac{Ymix}{Ymon}
$$

where $Ymix$ is the observed yield of species $i$ in mixture and $Ymon$ is the observed yield of species $i$ in monoculture expressed as a ratio ($Y_{gra(Mix)}$:$Y_{gra}$ and $Y_{leg(Mix)}$:$Y_{leg}$). Ratios that contribute to the analysis of overyielding: $Y_{gra}$:$Y_{leg}$ yield of grass vs legume in monoculture; $Y_{gra(Mix)}$:$Y_{leg(Mix)}$ yield of grass vs legume in mixture; $Y_{gra,leg}$:$\hat{Y}_{gra,leg}$ total yield in mixed cultivation compared with expected cumulative yield of constituent species according to monocultures; $Y_{gra(Mix)}$:$Y_{gra}$ yield of grass in mixture vs monoculture (RYT); $Y_{leg(Mix)}$:$Y_{leg}$ yield of legume in mixture vs monoculture (RYT). Where overyield was detected from calculations, we determined the response to be statistically significant where the mixed model results revealed that the factor 'cultivation' significantly influenced species yield (sPSA DAP 70), and we obtained multiple comparisons using the 'emmeans' package [54] within R [43].

## The relationship between species-specific growth rates and nutrient uptake

Principal components analysis (PCA) using "prcomp" with varimax rotation in R [43] was conducted to examine relationships between our image-derived growth (RGR) and DAP 70 nutrient parameters (Soil extractable N and P and shoot N and P concentrations). Assumptions of sampling independence, normality, linear relationships between variable pairs and

moderate correlations were confirmed. The optimal number of explanatory factors from each PCA was chosen based on eigenvalues greater than 1 [55]. Varimax rotation was applied so that information explained by one factor was independent of information in the other factors to achieve simple structure [56], and loadings <0.30 were not considered significant. **S1 Table in** S1 File details the eigenvalues and varimax loadings for all our principal components analyses.

## Results

### Absolute growth (smoothed projected shoot area on DAP 70) is correlated with aboveground biomass in mixed pasture cultivations

Aboveground biomass (DAP 70) was significantly correlated with sPSA (DAP 70) at the whole pot level ($r^2$ 0.77, $p < 0.001$, n = 120) with half-pot level correlation coefficients differing between grasses ($r^2$ 0.85, $p < 0.001$, n = 239) and legumes ($r^2$ 0.67, $p < 0.01$, n = 240; **S1a-S1c Fig in** S1 File). There were no differences in these relationships between plants grown in monoculture or mixture.

### RGR in grasses is driven by increased N availability and by presence of legumes, and in legumes by P availability and the presence of grasses

Significant treatment effects for RGR are shown in Table 1. RGR in grasses began higher than legumes and declined more rapidly, and growth ceased in grasses grown in monoculture without N after DAP 60 (Fig 3b). Between DAP 35–40 grass in the HNHP and HNLP treatments grew most rapidly regardless of cultivation, but between DAP 40–50, grasses in mixture under the HNHP and HNLP treatments achieved the highest RGR. Between DAP 60–70 a three-way interaction (cultivation by N by P) emerged. Here, RGR in grass under N addition were

**Table 1. P-values of the Wald F-statistics derived from linear mixed effects models for the main and interacting effects.**

| DAP | 35–40 | 40–50 | 50–60 | 60–70 |
|---|---|---|---|---|
| Treatment | RGR (kpixels day$^{-1}$) Grass | | | |
| C*N*P | 0.443 | 0.783 | 0.517 | **<0.001** |
| C*N | 0.127 | 0.183 | **<0.001** | nr |
| C*P | 0.704 | 0.354 | 0.715 | nr |
| N*P | 0.163 | 0.063 | 0.46 | nr |
| Cultivation | 0.201 | **<0.001** | nr | nr |
| Nitrogen | **<0.001** | **<0.001** | nr | nr |
| Phosphorus | 0.666 | 0.257 | 0.074 | nr |
| | RGR (day$^{-1}$) Legume | | | |
| C*N*P | 0.658 | 0.931 | **0.042** | **0.024** |
| C*N | 0.092 | 0.185 | nr | nr |
| C*P | 0.558 | **0.043** | nr | nr |
| N*P | 0.847 | 0.678 | nr | nr |
| Cultivation | 0.168 | nr | nr | nr |
| Nitrogen | **<0.001** | 0.073 | nr | nr |
| Phosphorus | 0.656 | nr | nr | nr |

Cultivation (C), nitrogen (N) and phosphorus (P) on the half-pot level trait RGR in grass and legume. Growth intervals are indicated by DAP (days after planting). Significant effects are indicated at (α = 0.05). nr indicates 'not reported' (suppression due to interaction effects).

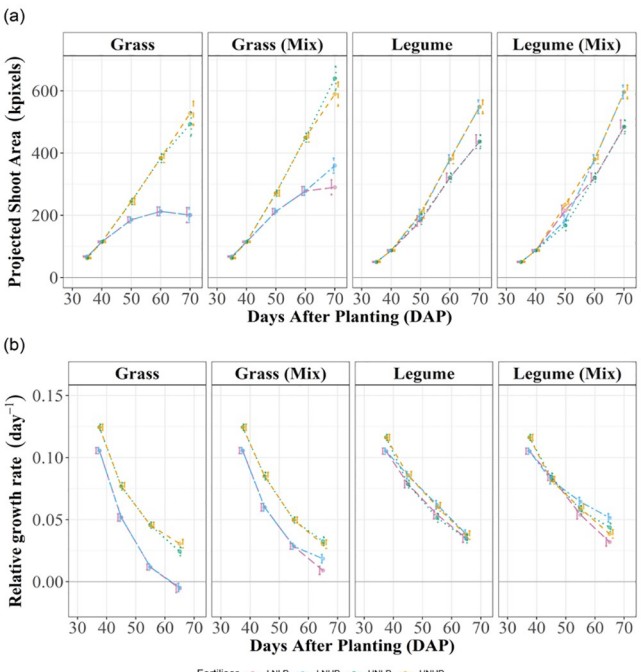

**Fig 3. Tables of estimated marginal means. a)** Estimated marginal means for AG (sPSA DAP 70, kpixels) and **b)** RGR (day$^{-1}$) by species (Eq 1), cultivation and nutrient treatment with half-least significant difference (5%) error bars. Non-overlapping error bars indicate significant differences. Purple = LNLP, Yellow = HNHP, Green = HNLP, Blue = LNHP.

comparable between cultivations, whereas under LN, grass RGR was greater in mixture. Grass RGR under LN with P addition (LNHP) was higher in mixture compared to monoculture.

Initial RGR in legumes (DAP 35–40) was increased by N addition (HNLP and HNHP treatments), but this effect did not persist for the remainder of the experiment (Fig 3b). Between DAP 40–60, RGR was increased by P addition in monocultures only. Between DAP 60–70, legume RGR was unaffected by nutrient treatment in monoculture but was altered in mixture where legumes benefited most under LNHP conditions. The lowest legume RGR was associated with N and P co-limitation in the LNLP mixed treatment.

## Grasses and legumes contribute to overyield in all nutrient treatments

As species in our study were separated aboveground with a divider, we discuss interspecific interactions from the standpoint that belowground interactions were occurring. Overyield (sPSA DAP 70) was present in all nutrient treatments, supporting the widely reported benefit of mixed grass-legume cultivations on total pasture growth (Fig 3a, Table 2; p<0.05). The ratio $Y_{gra,leg}$:$\hat{Y}_{gra,leg}$ reveals the effect of mixed cultivation on absolute growth (AG). At the whole pot level, productivity in mixtures increased in all treatments. On average, the LNHP treatment increased by 27%, followed by LNLP which increased 21%, then HNLP which increased 20% and finally HNHP which was 10% more productive in mixture (Table 2; p<0.05). The ratios $Y_{gra(Mix)}$:$Y_{gra}$ and $Y_{leg(Mix)}$:$Y_{leg}$ determined individual species yield expected in mixtures compared with those observed in monocultures (Table 2). Overyield was contributed to by both grasses and legumes.

Grass overyield was dependent on N or P addition, but not the two in combination (p<0.001). Grass under LNLP achieved 44% more biomass in mixture (p<0.001); in HNLP

**Table 2. Estimated marginal means of yield (sPSA DAP 70) and calculations of overyield (RYT, Eq 2), between grasses (Gra) and legumes (Leg) across nutrient treatments.**

| Fertiliser | sPSA± SE Gra | sPSA± SE Gra (Mix) | sPSA± SE Leg | sPSA± SE Leg (Mix) | $Y_{gra}$:$Y_{leg}$ | $Y_{gra(Mix)}$:$Y_{leg(Mix)}$ | $Y_{gra,leg}$:$\hat{Y}_{gra,leg}$ | $Y_{gra(Mix)}$:$Y_{gra}$ (RYT) | $Y_{leg(Mix)}$:$Y_{leg}$ (RYT) |
|---|---|---|---|---|---|---|---|---|---|
| HNHP | 527.9 (24.4) [de] | 589.5 (30.7) [ef] | 548.6 (15.5) [c] | 596.3 (18.0) [d] | 0.96 | 0.98 | 1.10 | 1.11 | 1.08 |
| LNHP | 200.8 (16.8) [a] | 359.4 (17.9) [c] | 548.6 (15.5) [c] | 596.3 (18.0) [d] | 0.37 | 0.60 | 1.27 | 1.78 | 1.08 |
| HNLP | 493.3 (24.4) [d] | 639.9 (30.7) [f] | 436.6 (15.5) [a] | 484.2 (18.0) [b] | 1.12 | 1.32 | 1.20 | 1.29 | 1.10 |
| LNLP | 200.4 (16.8) [a] | 289.8 (17.9) [b] | 436.6 (15.5) [a] | 484.2 (18.0) [b] | 0.45 | 0.59 | 1.21 | 1.44 | 1.10 |

$Y_{gra}$:$Y_{leg}$ = yield of grass *vs* legume in monoculture; $Y_{gra(Mix)}$:$Y_{leg(Mix)}$ yield of grass *vs* legume in mixture; $Y_{gra,leg}$:$\hat{Y}_{gra,leg}$ yield in mixed cultivation compared with expected cumulative yield of constituent species according to calculations monocultures; $Y_{gra(Mix)}$:$Y_{gra}$ yield of grass in mixture *vs* monoculture (RYT); $Y_{leg(Mix)}$:$Y_{leg}$ yield of legume in mixture *vs* monoculture (RYT). All means (kpixels) and ratios were calculated from sPSA (DAP 70) values. Different letters represent significant treatment (nutrient and cultivation) differences at $\alpha = 0.05$ within species.

mixture grass achieved a 29% increase in biomass compared with monoculture (p<0.001). HNHP treatment overyield was not detected for grasses (Fig 3a; Table 2). The highest overyield was detected in the LNHP treatment, where grass was 78% more productive in mixture than in monoculture (p<0.001). The reliance of grasses on mineral N or by co-occurring legumes, was further evidenced by the fact that in LNHP and LNLP, grass monocultures could not maintain growth after DAP 60 (Fig 3a).

Legumes overyielded in all mixtures (Table 2), the highest attributed to P addition treatments (Fig 3a). Growth in mixed HP treatments (HNHP, LNHP) was 8% higher compared with monoculture, and although LP treatments (HNLP, LNLP) achieved lower yields than the HP treatments they were 10% more productive in mixture compared with monoculture.

## Nitrogen and phosphorus interact to influence shoot nutrient concentrations in grasses and legumes between cultivations

Grasses increased shoot N (mg g$^{-1}$) in response to N addition and being grown in mixture, where the effect of N addition was augmented (p<0.001, Table 3). Grass shoot N under HNLP and HNHP treatments was 32% higher in mixture, and in LNLP and LNHP treatments was

**Table 3. Estimated marginal means of shoot nutrient concentrations (mg g$^{-1}$).**

| Fertiliser Treatment | Nut (mg g$^{-1}$) ± SE Gra | Nut (mg g$^{-1}$) ± SE Gra (Leg) | Nut (mg g$^{-1}$) ± SE Leg | Nut (mg g$^{-1}$) ± SE Leg (Gra) |
|---|---|---|---|---|
| | | **NITROGEN** | | |
| HNHP | 14.34 (0.5) [c] | 18.96 (0.7) [d] | 31.06 (0.8) [b] | 31.06 (0.8) [b] |
| LNHP | 7.00 (0.2) [a] | 9.18 (0.2) [b] | 32.58 (1.0) [b] | 32.58 (1.0) [b] |
| HNLP | 14.34 (0.5) [c] | 18.96 (0.2) [d] | 29.17 (0.8) [a] | 29.17 (0.8) [a] |
| LNLP | 7.00 (0.2) [a] | 9.18 (0.2) [b] | 26.87 (1.0) [a] | 26.87 (1.0) [a] |
| | | **PHOSPHORUS** | | |
| HNHP | 3.48 (0.1) [g] | 3.75 (0.1) [h] | 1.98 (0.07) [b] | 1.98 (0.07) [b] |
| LNHP | 3.09 (0.1) [e] | 3.36 (0.1) [f] | 1.98 (0.07) [b] | 1.98 (0.07) [b] |
| HNLP | 1.17 (0.1) [a] | 1.45 (0.1) [b] | 1.27 (0.07) [a] | 1.27 (0.07) [a] |
| LNLP | 1.84 (0.1) [c] | 2.12 (0.07) [d] | 1.27 (0.07) [a] | 1.27 (0.07) [a] |

Nutrient concentrations (mg g$^{-1}$) were measured in shoot biomass harvested on DAP 70 for Grass (Gra) and Legume (Leg). Different letters represent significant treatment (nutrient and cultivation) differences within species, at $\alpha = 0.05$.

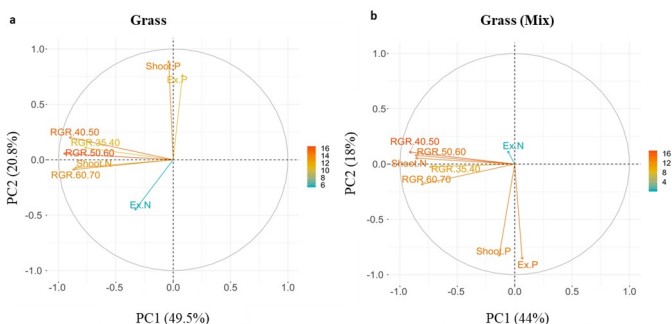

**Fig 4. PCA loading plot of the first two principal components for a) grasses in monoculture and b) grasses in mixture.** Gradient colour scales and arrow length represent the percentage of variation explained by each variable along the principal component. RGR 35–70 = Relative growth rate (kpixels day$^{-1}$), Ex P = Extractable P (mg g$^{-1}$), Ex N = Extractable N (mg g$^{-1}$), Shoot P = Shoot P (mg g$^{-1}$), Shoot N = Shoot N (mg g$^{-1}$).

31% higher. Legume shoot N (mg g$^{-1}$) was only increased by P addition, with no additional effect in mixture (p<0.001).

Shoot P (mg g$^{-1}$) in grass was influenced by an interaction between N and P (p<0.001), cultivation and P (p<0.01) and cultivation and N (p<0.01). Shoot P increased when N was added or in mixtures. Without N but with P addition (LNHP), grass shoot P was 8% higher in mixtures. The highest shoot P concentration was in grasses under the HNHP mixed treatment where it increased by 7% from monoculture. Shoot P (mg g$^{-1}$) in legumes was influenced by P status (p<0.001) being higher when P was added but was unaffected by cultivation with grasses (Table 3).

## Grass RGR parameters are related to shoot N in both cultivations, and legume RGR parameters diverge between monocultures and mixtures

Principal components analysis revealed differences in the relatedness of nine parameters pertaining to growth and nutrient use between grasses and legumes (**S2 and S3a-S3b Figs in** S1 File). Although DAP 35–70 RGR parameters were largely correlated in grasses, in legumes they were not and as such we decided to include all RGR parameters in our analysis to allow comparison between species.

Over the duration of the experiment, RGR in grass grown in both monoculture and mixture was strongly correlated with shoot N, and to a lesser degree, extractable N; it was uncorrelated (orthogonal) to shoot and extractable P concentrations (Fig 4a and 4b). Additionally, the correlation in RGR intervals suggests that RGR displayed consistency throughout the experiment for grasses.

Legume RGR in monoculture became increasingly correlated with nutrient availability, especially shoot P and extractable P, over the course of the experiment. Legume RGR in mixture showed a similar pattern, with a strong correlation to shoot and extractable P at the end of the experiment, but with a stronger correlation to extractable N and shoot N in the early stages of the experiment (Fig 5a and 5b).

## Discussion

### Growth strategy is determined by nutrient availability and community interactions

Facilitation of N uptake [21] and growth in grass-legume mixtures is the primary motivation for intercropping in pastoral agriculture. N and P fertilization can have varying effects on this

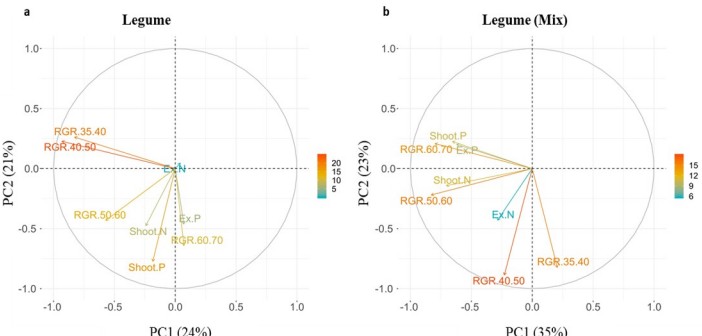

**Fig 5. PCA loading plot of the first two principal components for a) legumes in monoculture and b) legumes in mixture.** Gradient colour scales and arrow length represent the percentage of variation explained by each variable along the principal component. RGR 35–70 = Relative growth rate (kpixels day$^{-1}$), Ex P = Extractable P (mg g$^{-1}$), Ex N = Extractable N (mg g$^{-1}$), Shoot P = Shoot P (mg g$^{-1}$), Shoot N = Shoot N (mg g$^{-1}$).

relationship [57], as well as the length of time that species interact under particular nutrient conditions [58]. We successfully measured growth facilitation using HTP, revealing a yield benefit for both grasses and legumes (Fig 3a). Grasses displayed a faster more acquisitive growth strategy, while legumes reduced growth rate, increased AG but maintained shoot nutrient concentrations (Fig 3a and 3b, Tables 1–3). Variation in species-specific growth strategies can shape resource use and primary productivity in agricultural systems as they determine requirements for N and P [59]. Using HTP, we accurately detected interspecific differences in response to nutrients, revealing higher AG and faster RGR in grasses responding to N, and a slower, more conservative growth strategy in legumes responding to P (Fig 3b). RGR in grasses was consistently related to shoot N concentrations in both cultivations, whilst in legumes, nutrient effects on growth were only detected towards the mid-to-late stages and were more pronounced in mixture (DAP 50–70), (Figs 4 and 5).

## Mixed cultivation increases growth in both species and augments N and P uptake in grasses

Combining grass and legume is beneficial for pasture yield as legumes can increase grass growth via transfer of fixed N [60], and for pasture quality as the facilitative interaction can increase nutrient uptake and storage [16, 61, 62]. In addition to the facilitative benefit mixed cultivation presented for grasses, our data also revealed a clear growth benefit for legumes. Whilst not commonly discussed that grasses 'facilitate' legumes, grasses are demonstrated to prevent N leaching *via* their root structures [63] and to trigger up-regulation of BNF to benefit the legume [18, 58, 64]. In our study, grasses increased their shoot N and P concentrations in mixtures, and despite a growth increase in mixture, legume N and P concentrations remained constant. Robust, positive relationships between foliar N concentrations, RGR and yield have been reported in grasses [65], and the positive response of grass to N fertilization has been physiologically related to its fast RGR [66], which strongly determines a plant's metabolic requirement for N and P [30]. In our study, increased RGR and foliar N concentrations in grasses following N addition with an augmented effect in mixture, suggested both a facilitative effect of legumes on grass growth [21] and the presence of an exploitative nutrient uptake strategy in grasses [66–68]. Our observed increases to shoot P concentrations (mg g$^{-1}$) in grasses under N addition or in mixture, coupled with increased growth, provides support for the well-known theory of an increased P requirement in faster growing organisms [30, 69].

## Legumes alter their growth in the presence of grasses but maintain consistent shoot nutrient concentrations

RGR in legumes appeared to differ between early-to-mid stage growth (DAP 35–50) that was significantly correlated with soil N availability in mixtures, and late stage growth (DAP 50–70) was not correlated with soil N. We suggest that during this early stage growth related more to mineral N than biologically fixed N, as legume nodules may not yet have properly developed [44, 45]. The mid-to-late legume growth stage (DAP 50–70), was associated with soil P and shoot N and P concentrations in both cultivations, but despite legume RGR slowing in mixture, there was no cultivation difference in final shoot nutrient concentrations. The legume RGR being more strongly affected by P availability in mixture towards the end of the experiment is likely owing to the prolonged interaction with grasses which may have up-regulated biological nitrogen fixation [27, 70]. A more conservative growth and nutrient uptake strategy in legumes that is mediated by community interactions and facilitates increased growth [20, 67] is positive for increasing overall forage quality and productivity in mixed pastures [16, 21, 62, 71].

## The potential value of high-throughput phenotyping for mixed cultivations

High throughput phenotyping continues to develop as a promising technique to replace at least some of the traditional approaches to plant functional trait assessment [35, 36]. As our study is one of the first to examine mixed species, there was some expected variability in the characterization of projected shoot area measured against harvested biomass between species having different growth forms. In previous monoculture studies, correlations between projected shoot area obtained from RGB images and harvested plant biomass have generally been demonstrated as strong, with variations reported in relation to growth stage and plant height [72, 73] and plant functional type (grass *vs* legume) [74]. In the latter case, however, only canopy height was assessed, sensing was remotely obtained, and images were taken from a primarily top view. We believe that the combined use of top and side-view cameras may have reduced some of this error in our study, leading to relatively strong correlations ($r^2$ ranging from 0.67 to 0.85) despite the interspecific differences. Nevertheless, the future capacity of HTP to inform agricultural management will continue to rely on robust and repeated calibrations in both controlled and field-based systems, and validation of acquired data in systems of interest.

## Conclusion

Pastoral agriculture is a multi-billion-dollar industry that currently relies heavily on intensive fertilization for effective agricultural production. Novel sampling techniques such as image-based phenotyping have the potential to revolutionize our understanding of plant growth dynamics, thereby improving our ability to reduce reliance on mineral fertilizers. In addition, this dataset, calibrated under highly controlled conditions contributes to a more comprehensive understanding of plant-plant interactions in common agricultural systems, providing a platform for future testing in field-based systems.

## Supporting information

**S1 File.**
(DOCX)

## Acknowledgments

The authors wish to acknowledge the invaluable contributions of APPF technical staff, in particular Lidia Mischis, Fiona Groskreutz and Nicole Bond who worked tirelessly to ensure that this experiment was successful. We thank Dr Guntur Tanjung and George Sainsbury for rapid and accurate image analyses throughout the experiment. Additionally, we acknowledge the incredible assistance of Dr Krista Plett, Johanna Wong and Emi Stuart without whom the final harvest would have been far more arduous.

## Author Contributions

**Conceptualization:** Kirsten Rae Ball, Sally Anne Power, Sarah Woodin, Elise Pendall.

**Data curation:** Kirsten Rae Ball, Chris Brien, Nathaniel Jewell.

**Formal analysis:** Kirsten Rae Ball, Chris Brien, Nathaniel Jewell.

**Funding acquisition:** Kirsten Rae Ball.

**Methodology:** Kirsten Rae Ball, Bettina Berger, Elise Pendall.

**Project administration:** Kirsten Rae Ball.

**Resources:** Bettina Berger.

**Supervision:** Kirsten Rae Ball, Sally Anne Power, Sarah Woodin, Bettina Berger, Elise Pendall.

**Visualization:** Kirsten Rae Ball.

**Writing – original draft:** Kirsten Rae Ball.

**Writing – review & editing:** Kirsten Rae Ball, Sally Anne Power, Sarah Woodin, Bettina Berger, Elise Pendall.

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
