## [Decision Letter · Decision Letter 0]

25 Aug 2020

PONE-D-20-20899

High-throughput, image-based phenotyping reveals nutrient-dependent growth facilitation in a grass – legume mixture

PLOS ONE

Dear Dr. Ball,

Thank you for submitting your manuscript to PLOS ONE. After careful consideration, we feel that it has merit but does not fully meet PLOS ONE’s publication criteria as it currently stands. Therefore, we invite you to submit a revised version of the manuscript that addresses the points raised during the review process.

We look forward to receiving your revised manuscript.

Kind regards,

Roberto Papa, PhD

Academic Editor

PLOS ONE

Journal Requirements:

'This project was supported in part by a Postgraduate Internship Award by the Australian Plant Phenomics Facility (APPF) awarded to Kirsten Ball. The APPF is funded by the Australian Government under the National Collaborative Research Infrastructure Strategy (NCRIS).'

'This project was supported in part by a Postgraduate Internship Award by the Australian Plant Phenomics Facility (APPF) awarded to KB.

https://www.plantphenomics.org.au/

Staff at the APPF (BB, CB, NJ) assisted in experimental design, statistical analyses and manuscript development.'

5. Please include captions for your Supporting Information files at the end of your manuscript, and update any in-text citations to match accordingly. Please see our Supporting Information guidelines for more information: http://journals.plos.org/plosone/s/supporting-information

Reviewers' comments:

Reviewer's Responses to Questions

**Comments to the Author**

1. Is the manuscript technically sound, and do the data support the conclusions?

Reviewer #1: Yes

Reviewer #2: Yes

2. Has the statistical analysis been performed appropriately and rigorously? 

Reviewer #1: Yes

Reviewer #2: N/A

3. Have the authors made all data underlying the findings in their manuscript fully available?

Reviewer #1: No

Reviewer #2: Yes

4. Is the manuscript presented in an intelligible fashion and written in standard English?

Reviewer #1: Yes

Reviewer #2: Yes

5. Review Comments to the Author

Reviewer #1: It was not possible to find a URL/accession or a number/DOIs where it was possible to download the data, unlike what was declared.

I would like to bring to the attention of the author some passages: row (64-65-66), reference is made to the types of interaction between plants suggesting that there was a sequence between them, which is not the case.

In line (102), reference is made to the objectives of the study using a past tense "were", usually a tense is used at present.

In row (182), reference is made to the addition of 99 mg N and 11 of P with a ratio of (6: 1), this ratio is incorrect.

In line (241, 242), reference is made to the methodology only through the numerical label of the bibliography, in my opinion I do not find it properly correct.

Furthermore, I would like to bring to the attention of the author the fact that in some cases the double quotation in the same sentence is separated by a comma (line 87), while in other cases it is separated by a hyphen (line 70).

Overall in my opinion it is well done and the concepts are well expressed and connected.

Reviewer #2: Overall, the manuscript is well done and highlights important issues, which could be used in further studies on mixtures and are applicable also on other species. Moreover, it underlines hight-throughtput phenotyping potential for studying crop growth patterns and facilitation in different conditions (i.e. nutrient availability) and its needs for careful calibration and validation. It only needs a minor revision in terms of sintax and lexical form. Result discussion is very well done and conclusions are strongly supported by data.

I put some advices, comments and corrections in the following lines.

17: delete “To our knowledge”

25-28: focus attention on results achieved in the mixture, rather than indicate species favourite nutritional conditions, that are largely known

53-56: please, better define this point, it’s not so clear

60: biological nitrogen fixation or biological N2 fixation

84: delete “Regardless of the measurement”

88: delete comma after “rapid”

98: delete “Regardless of the technique”

111 and 116: biological nitrogen fixation or biological N2 fixation

123: I suggest “Experimental design and plant material”; this section is not so clear; I put some suggestion in the following lines.

124-126: introduce better the experimental site, indicating GPS coordinates, specify that the experiment was conducted in a greenhouse using Lemnatec Imaging System and move the info about the images in section 2.6

127-130: it could be better to modify this sentence. I propose “The experiment investigated the effects of four fertiliser treatments derived by factorial combination of two levels of nitrogen (LN and HN) and two levels of phosphorus (LP and HP) on one grass (Phalaris aquatica) and one legume (Trifolium vesiculosum) pasture species grown in monoculture and mixture.”

130-131: it could be better to modify this sentence. I propose “The twelve treatments were arranged in a latinized, resolved incomplete-block design with ten replicates, for a total of 120 pots.”

139-143: I suggest to reorganize this part, adding in brackets the nutrient combinations (LNLP, HNHP, HNLP and LNHP) and deleting the reference to the experimental design, which you will move in the main text. I suggest: “Within each replicate (indicated by the solid red box) the four nutrient combinations (LNLP, HNHP, HNLP and LNHP) were assigned to 4 main units arranged in a grid according to the experimental design. The three species combinations [grass only (Gra), legume only (Leg) and mixture (Gra Leg)] were randomized to the three consecutive pots within each main unit. Colored blocks indicate 1 pot containing two halves.”

145: I suggest to rename this part in “Growing conditions”

146: please add a sentence to specify growth condition in general, in order to better introduce the experimental conditions and parameters description.

149-151: please, improve the sentence

174: I suggest “Fertilisation treatments”

178-179: please, adjust the sentence, including punctuation and brackets

182: ratio 9:1

182-183: adjust punctuation and brackets

187: indicate in which table number

361-362: add punctuation and adjust brackets

367: you can add the reference on equation 1

375, 387, 391, 403: correct the figure number

376, 380: I guess you should better introduce the use of ratios in the main text, maybe in the section 2.10

376-382: please, improve this part

393-400: caption is not so clear; you could improve it moving some informations in the main text and reorganizing the text in order to promote immediate understanding

422-424: explain in the caption text what Gra (Leg) and Leg (Gra) mean

443, 454: add a bracket after “shoot N”

456: in all this section, please add tables and figures references.

464: delete “in mixture”

479: add comma after “grasses”

528: “conclusion” should have a separate paragraph

6. PLOS authors have the option to publish the peer review history of their article (what does this mean?). If published, this will include your full peer review and any attached files.

Reviewer #1: **Yes: **Papalini Simone

Reviewer #2: No

---

## [Author Response · Author response to Decision Letter 0]

9 Sep 2020

Dear Dr Papa,

We enclose herewith our revised manuscript entitled ‘High-throughput, image-based phenotyping reveals nutrient-dependent growth facilitation in a grass – legume mixture’ for consideration for publication in the “Plant Phenomics and Precision Agriculture” special edition. 

We have revised the manuscript accounting for PLOS ONE’s style requirements as requested and have provided repository information for the data used in the manuscript. At present this is a conditional DOI (10.25909/12895121) which will be linked to the article upon publication. Presently, the data can be accessed via this private link: https://figshare.com/s/99e05c190be6cb416164

The updated financial disclosure which has been removed from the acknowledgements section is as follows: “The Australian Plant Phenomics Facility received grant funding from the Australian Government through the National Collaborative Research Infrastructure Strategy (NCRIS). KB received a Postgraduate Internship Award from the Australian Plant Phenomics Facility towards the completion of this project”.

We wish to sincerely thank the reviewers for their valuable assistance in refining the manuscript for publication and provide responses to their comments herein. We hope that you find these responses satisfactory, and we look forward to positive news of our article’s acceptance.

Yours sincerely,

Kirsten Ball (on behalf of all co-authors)

 

Reviewer #1: 

It was not possible to find a URL/accession or a number/DOIs where it was possible to download the data, unlike what was declared.

RESPONSE: Thank you for this comment, the associated data for the manuscript has now been placed in a repository and the provisional DOI is 10.25909/12895121, which once published will be linked to the article. Until the article is published, the data can be viewed via this private link: https://figshare.com/s/99e05c190be6cb416164

I would like to bring to the attention of the author some passages: row (64-65-66), reference is made to the types of interaction between plants suggesting that there was a sequence between them, which is not the case.

RESPONSE: Thank you for this suggestion. This syntax has now been changed to “Interspecific interactions between plants can be negative, in the case of competition for resources [22], neutral, where complementarity ensures that species do not compete for the same resources [11], and positive, where facilitation leads to higher performance of a species when grown in mixture than in monoculture [23]” removing the suggestion of sequence.

In line (102), reference is made to the objectives of the study using a past tense "were", usually a tense is used at present.

RESPONSE: Thank you. The word “were” was used not to denote tense but because there were multiple aims of the study, hence “the aims were….” We have now changed this to “This study used” to clarify.

In row (182), reference is made to the addition of 99 mg N and 11 of P with a ratio of (6: 1), this ratio is incorrect.

RESPONSE: Thank for pointing out this typographic error. It has now been rectified. 

In line (241, 242), reference is made to the methodology only through the numerical label of the bibliography, in my opinion I do not find it properly correct.

RESPONSE: The method employed has now been specified in the initial sentence and the following description modified to emphasize the steps involved in the method. These are now lines 248-250.

Furthermore, I would like to bring to the attention of the author the fact that in some cases the double quotation in the same sentence is separated by a comma (line 87), while in other cases it is separated by a hyphen (line 70).

RESPONSE: Where there are only 2 references in the citation a comma is used, in the case where there are more than 2, a hyphen is used. 

Overall in my opinion it is well done and the concepts are well expressed and connected.

Reviewer #2: 

Overall, the manuscript is well done and highlights important issues, which could be used in further studies on mixtures and are applicable also on other species. Moreover, it underlines hight-throughtput phenotyping potential for studying crop growth patterns and facilitation in different conditions (i.e. nutrient availability) and its needs for careful calibration and validation. It only needs a minor revision in terms of sintax and lexical form. Result discussion is very well done and conclusions are strongly supported by data.

I put some advices, comments and corrections in the following lines.

17: delete “To our knowledge”

RESPONSE: Completed.

25-28: focus attention on results achieved in the mixture, rather than indicate species favourite nutritional conditions, that are largely known

RESPONSE: Thank you for this comment. We have removed the single-species favored nutrient conditions from these lines and included only the interactions that occurred in mixtures.

53-56: please, better define this point, it’s not so clear

RESPONSE: This has been altered to make the point clearer “Plant growth strategies involve trade-offs between increasing both productivity and resource acquisition (interpreted herein as an acquisitive growth strategy), and reducing productivity and conserving resources (interpreted herein as a conservative growth strategy) [14].”

60: biological nitrogen fixation or biological N2 fixation

RESPONSE: Completed 

84: delete “Regardless of the measurement”

RESPONSE: Completed.

88: delete comma after “rapid”

RESPONSE: Completed

98: delete “Regardless of the technique”

RESPONSE: Completed

111 and 116: biological nitrogen fixation or biological N2 fixation

RESPONSE: Completed

123: I suggest “Experimental design and plant material”; this section is not so clear; I put some suggestion in the following lines.

RESPONSE: Thank you. We have now changed this to “Experimental design and species”

124-126: introduce better the experimental site, indicating GPS coordinates, specify that the experiment was conducted in a greenhouse using Lemnatec Imaging System and move the info about the images in section 2.6

RESPONSE: We have revised this sentence accordingly: “Our experiment was conducted between the 24th July 2018 and 5th October 2018 at the Australian Plant Phenomics Facility at the University of Adelaide (-34.971298, 138.639627) in a Lemnatec Imaging System”

127-130: it could be better to modify this sentence. I propose “The experiment investigated the effects of four fertiliser treatments derived by factorial combination of two levels of nitrogen (LN and HN) and two levels of phosphorus (LP and HP) on one grass (Phalaris aquatica) and one legume (Trifolium vesiculosum) pasture species grown in monoculture and mixture.”

RESPONSE: We have altered the sentence according to the reviewer’s suggestion. 

130-131: it could be better to modify this sentence. I propose “The twelve treatments were arranged in a latinized, resolved incomplete-block design with ten replicates, for a total of 120 pots.”

RESPONSE: We have altered the sentence according to the reviewers suggestion.

139-143: I suggest to reorganize this part, adding in brackets the nutrient combinations (LNLP, HNHP, HNLP and LNHP) and deleting the reference to the experimental design, which you will move in the main text. I suggest: “Within each replicate (indicated by the solid red box) the four nutrient combinations (LNLP, HNHP, HNLP and LNHP) were assigned to 4 main units arranged in a grid according to the experimental design. The three species combinations [grass only (Gra), legume only (Leg) and mixture (Gra Leg)] were randomized to the three consecutive pots within each main unit. Colored blocks indicate 1 pot containing two halves.”

RESPONSE: We have altered the sentence according to the reviewer’s suggestion.

145: I suggest to rename this part in “Growing conditions”

RESPONSE: We have altered the header according to the reviewers suggestion.

146: please add a sentence to specify growth condition in general, in order to better introduce the experimental conditions and parameters description.

RESPONSE: Thank you or this suggestion however it is the authors belief that the methods section is already quite lengthy, and much attention has already been paid to demonstrating growth conditions using both text and figures. We have therefore not included more explanation as suggested. 

149-151: please, improve the sentence

RESPONSE: We have now improved the sentence accordingly:” The plants were physically separated aboveground by a white plastic divider, set 5 cm into the soil and oriented north–south on the conveyor system in the greenhouse for consistency with respect to imaging orientation and solar exposure. The divider allowed intermingling of roots belowground (Fig 2). 

174: I suggest “Fertilisation treatments”

RESPONSE: We have changed the header per the reviewer’s suggestion

178-179: please, adjust the sentence, including punctuation and brackets

RESPONSE: Thank you. We have now separated these sentences for the readers ease.

182: ratio 9:1

RESPONSE: Altered. Thank you.

182-183: adjust punctuation and brackets

RESPONSE: These alterations have been made, and brackets added around the units of measurement. 

187: indicate in which table number

RESPONSE: Thank you for this suggestion however this information is not in tabular form but is present and clearly headed in the supplement. 

361-362: add punctuation and adjust brackets

RESPONSE: These alterations have been made

367: you can add the reference on equation 1

RESPONSE: Thank you. This has been added. 

375, 387, 391, 403: correct the figure number

RESPONSE: These have been corrected. 

376, 380: I guess you should better introduce the use of ratios in the main text, maybe in the section 2.10

RESPONSE: We have now included this part in the section “Calculating overyielding from sPSA DAP 70 values”: We have altered the text accordingly: “Ymix is the observed yield of species i in mixture and Ymon is the observed yield of species i in monoculture expressed as a ratio (Ygra(Mix):Ygra and Yleg(Mix):Yleg).

376-382: please, improve this part

RESPONSE: We have broken down the sections and improved the structure accordingly: “At the whole pot level, productivity in mixtures increased in all treatments. On average, the LNHP treatment increased by 27%, followed by LNLP which increased 21%, then HNLP which increased 20% and finally HNHP which was 10% more productive in mixture” 

393-400: caption is not so clear; you could improve it moving some informations in the main text and reorganizing the text in order to promote immediate understanding

RESPONSE: Thank you for this suggestion. We have now moved the larger proportion of the figure caption up into the section “Calculations of overyielding” and made the caption text clearer. 

422-424: explain in the caption text what Gra (Leg) and Leg (Gra) mean

RESPONSE: This has now been added to the figure caption.

443, 454: add a bracket after “shoot N”

RESPONSE: Added. Thank you.

456: in all this section, please add tables and figures references.

RESPONSE: Thank you for this valuable suggestion, we have now added all table and figure references. 

464: delete “in mixture”

RESPONSE:

479: add comma after “grasses”

RESPONSE: Altered.

528: “conclusion” should have a separate paragraph

RESPONSE: Thank you. This section is its own separated paragraph with header “Conclusion”

---

## [Editor Report · Decision Letter 1]

11 Sep 2020

High-throughput, image-based phenotyping reveals nutrient-dependent growth facilitation in a grass – legume mixture

PONE-D-20-20899R1

Dear Dr. Ball,

We’re pleased to inform you that your manuscript has been judged scientifically suitable for publication and will be formally accepted for publication once it meets all outstanding technical requirements.

Kind regards,

Roberto Papa, PhD

Academic Editor

PLOS ONE
---

## [Editor Report · Acceptance letter]

28 Sep 2020

PONE-D-20-20899R1 

High-throughput, image-based phenotyping reveals nutrient-dependent growth facilitation in a grass - legume mixture 

Dear Dr. Ball:

I'm pleased to inform you that your manuscript has been deemed suitable for publication in PLOS ONE. Congratulations! Your manuscript is now with our production department. 

Kind regards, 

on behalf of

Prof. Roberto Papa 

Academic Editor

PLOS ONE